# Understanding Tourist Behavioural Intention When Faced with Smog Pollution: Focus on International Tourists to Beijing

**DOI:** 10.3390/ijerph18147262

**Published:** 2021-07-07

**Authors:** Wenjia-Jasmine Ruan, Junjae Lee, Hakjun Song

**Affiliations:** 1School of Tourism Management, Sun Yat-sen University, Zhuhai 519000, China; ruanwj6@mail.sysu.edu.cn; 2Department of Convention and Hotel Management, Hannam University, Daejeon 34430, Korea; hmanjj@hanmail.net; 3Department of Hotel and Service Management, Pai Chai University, Daejeon 35345, Korea

**Keywords:** smog pollution, inbound tourism, behavioural intention, mass media, MGB

## Abstract

This study examines the behavioural intentions of international tourists travelling to Beijing when faced with smog pollution. An extended MGB (model of goal-directed behaviour) was employed as the theoretical framework by integrating mass-media effect and perception of smog. The role of mass-media effect and perception of smog were considered as new variables in the international tourist’s decision-making process for travel to Beijing. Structural equation modelling (SEM) was employed to identify the structural relationships among research variables. Our research results showed a strong correlation between positive anticipated emotion and desire. The mass-media effect is a significant (direct) predictor of both the perception of smog and behavioural intention. The Chinese government could attach great importance to the mass-media effect to reduce the negative impact caused by smog pollution on inbound tourism.

## 1. Introduction

Environmental quality has gained increasing recognition for its ability to enhance a destination’s competitive advantage. It is also important as a reference index for destination sustainability, as the environment plays a prime consideration in an individual’s decision to travel [1,2]. Smog pollution is a prominent influence on human activity and behaviour because of its higher visibility and severity compared to other types of environmental pollution, such as water pollution, improper waste management, and soil pollution [3,4]. For some developing countries, the deteriorating air quality has become the main reason for the decreasing number of tourist arrivals and tourism revenue, adversely affecting the economies of these countries [5]. For example, China’s attractiveness as a tourist destination was greatly weakened in 2013 due to severe “smog” (“wumai” in Chinese) [6]; international tourist arrivals to Beijing totalled 26.29 million in 2013, down 3.3% from the previous year [7]. The current widespread air pollution issue in China was mainly responsible for the decline in inbound tourist market [8]. Therefore, the State Council of China announced the Air Pollution Prevention and Control Plan on 10 September 2013 with the purpose of controlling polluted air emissions and further improving air quality within five years. Until 2017, the air quality of China improved greatly compared to that of 2012. However, the annual concentration of particulate matter in Beijing, namely PM_2.5_ (58 μg/m^3^) and PM_10_ (84 μg/m^3^), continues to exceed WHO guidelines (PM_2.5_: 10 μg/m^3^; PM_10_: 20 μg/m^3^) and poses substantial threats to human health (See Figure 1). Although air quality in China has been increasing with the Chinese government’s continuous efforts, there is still a long way to go, as the improvement of China’s environment quality is not stable. Thus, smog pollution in China seems to still be a constraint on international tourist arrivals.

The mass media have accelerated the spread of news and reports about China’s smog around the world. The Google search index of China’s smog pollution has increased dramatically since January 2013, which coincided with severe smog pollution in China (See Figure 2). Among enormous reports, a recent study in the lancet estimated that in 2017, 1.24 million people died from air pollution in China [9]. This study result was cited by the U.S. Embassy in Georgia, reporting that air pollution in China harms its citizens and even the world. Furthermore, WHO labelled air pollution in China a health crisis, and according to the World Economic Forum 2019, China’s overall tourism competitiveness ranks 13th among 140 countries, whereas its environmental sustainability index ranks 120th, with air pollution ranking 137th [10]. The adverse news and reports on China’s smog pollution are the primary reasons for the damage to the inbound tourist market [8]. As argued by Li, Pearce, Morrison, and Wu [11], the media coverage of smog pollution may destroy a city’s established tourism image, possibly resulting in a shrinking tourism market and reduced revenue. Therefore, we can imply that smog pollution has become a major concern for international tourists visiting China, and mass media may be an important factor that can stir or strengthen public awareness of China’s smog pollution.

According to Murukutla, Negi, Puri, Mullin, and Onyon [12], mass-media coverage has a major ability to raise public’s risk perception of smog pollution; they also emphasized the potential achievements of mass-media advocacy on public and policy engagement towards air-quality governance. Although smog pollution has been recognized as a critical constrain factor for tourist arrivals both in China’s domestic and inbound tourism [13,14,15,16], there is limited literature devoted to explaining or predicting the role of mass-media effect in tourists’ decision-making process when considering the impact of smog pollution [14]. Faced with the decline in China’s inbound tourism, it is imperative to get a comprehensive understanding of the behavioural intention of international tourists considering the threat of smog pollution. The model of goal-directed behaviour (MGB) is an advanced theoretical framework that considers emotion and prior experience together to understand individual human behaviour. This study attempts to answer the following research questions under the guidance of the MGB: (1) What is the role of mass-media effect and perception of smog in the decision-making process of international tourists of Beijing? (2) Which variables are more important in behavioural intentions? (3) Can the EMGB (extended MGB), developed by including two new constructs (mass-media effect and perception of smog), be applied to predict behavioural intentions of international tourists?

## 2. Theoretical Background 

### 2.1. Smog Pollution and Tourism

In 1905, Dr. Henry coined the term “smog”, smoke and fog, to describe the smoky fog produced in great cities [17]. Smog pollution has been considered to be a leading cause of human mortality. According to the UN Environment, one-ninth of the world’s population has died prematurely due to smog pollution. Besides debilitating health effects, severe smog is also assumed to impact tourism destinations negatively [18]. Long-lasting and severe smog pollution has weakened the attractiveness and competitiveness of tourist destinations and hindered its sustainability. Some studies have tried to quantify the economic losses caused by smog pollution to the tourism industry, including testing the negative correlations between tourist arrival rate/tourism revenue and air quality index [19] and the negative relationship between smog pollution and inbound tourism demand [14]. Tang, Yuan, Ramos, and Sriboonchitta [14] empirically determined that smog pollution will have a negative impact on international tourists’ visiting intentions in the long term. Zhang, Ren, Zhang, and Zhang [20] believed that there is a correlation between tourism development and smog pollution. Specifically, smog pollution, which affects the air quality of a city, will hinder the development of tourism.

### 2.2. Smog Pollution in CHINA 

China is facing a chronic and complicated problem due to smog pollution: negative impact on human lifespan, inbound tourism development, and national image. According to OECD (Organization for Economic Co-operation and Development), up to 9 million people worldwide will die prematurely by 2060 based on the current smog pollution level, with China suffering the worst number of fatalities [21]. Notably, this was supported by satellite data showing that Beijing is one of the primary victims of environmental problems caused by China’s spectacular economic growth in recent decades. 

Smog pollution is widely considered to be a primary reason for the decline in the number of China’s inbound tourists in recent years [8]. Xu and Reed [16] stated that individual perception of the severity of China’s smog pollution can impede inbound tourism. To control smog pollution, the Chinese State Council released the Air Pollution Prevention Plan on 10 September 2013 [22]. Although smog pollution has had been gradually controlled since 2013, only 46.6% of China’s cities have reached the air quality standard of WHO, whereas 53.4% cities exceeded the standard according to the 2020 China Ecological Environment Bulletin, implying that China’s smog pollution is still a pressing issue to be solved.

### 2.3. Mass Media and Smog Pollution 

The mass media plays an important role in information dissemination, public opinion guidance, and persuasion [23]. Mass media not only supplies people with factual information about a specific issue but also gravitates public attention towards a particular topic [24]. Lippmann [25] asserted that the news media is a primary source of images about the larger world in people’s minds. People’s understanding and impression of the world are largely based on the information delivered by the news media. In China, news media plays a vital role in educating the public about the health threats of smog pollution [26].

Media coverage and publicity are the main communication channels to arouse public awareness of smog pollution [27]. Hong, Kim, and Xiong [28] stated that media exposure could positively enhance public disaster risk awareness and emergency preparedness behaviour. Contrarily, negative information from mass media may elevate a reader’s anxiety and negative emotions towards such a threat [29]. Smog in China is known rapidly worldwide due to extensive global media coverage [30]. In November 2010, the U.S. embassy in Beijing tweeted about Beijing smog on its official Twitter account, saying that the concentration of PM_2.5_ was above 500, indicating that the air quality of Beijing was “crazy bad”. Subsequently, the U.S. embassy’s reading was widely retweeted. Individuals were advised to avoid outdoor activities to reduce the possibility of exposure to smog or wear a mask when going out [31]. The large-scale media coverage raised public health awareness and stimulated discussion on smog [26,27]. Exposure to pollution-related reports or documents amplified risk perception, which was also influential in motivating people to search for more information of smog pollution and adopt preventive behaviours [32]. It is conceivable that mass-media coverage would increase the risk perception of potential inbound tourists and further influence the image of the tourist’s destination and his/her intention of visiting China [13].

### 2.4. Tourist’s Behavioural Intention under Air Pollution 

Severe smog is assumed to threaten the sustainable development of tourism destinations by jeopardizing their attractiveness and competitiveness [18]. Individuals mainly respond to air pollution with avoidant attitudes or by showing negative perception. For onsite tourists, Cheng, Zhang, and Fu [33] found that individuals prefer to avoid exposure to polluted air when travelling. Li et al. [11] indicated that Chinese domestic visitors to Beijing had a strong negative perception of smog, which is likely to significantly reduce visitor satisfaction of Beijing travel. It was also revealed that a tourist’s perception of health risk caused by smog may shorten his/her stay period and even lead to his/her disloyalty [34]. For potential tourists, air pollution was proven to be a crucial factor that will suppress China’s international tourism. Specifically, Becken, Jin, Zhang, and Gao [13] revealed that prospective international travellers’ cognitive and affective perceptions about air quality negatively influenced China’s destination image and tourists’ corresponding visit intentions. Xu and Reed [16] assessed the impact of pollution on potential international tourists and reported that perceived pollution can impede inbound tourism. Recently, Ruan, Kang, and Song [35] pointed out that international tourists will avoid travelling to China if they perceive a severity of air pollution in China.

### 2.5. MGB and Its Application in Tourism Context

The MGB is an advanced theoretical framework to explain and predict individual behaviour. It addresses the limitation of the Theory of Planned Behaviour (TPB) by integrating emotional factors, past behaviour, and desire [36]. Perugini and Bagozzi [36] concluded that MGB will help in advancing scientific understanding of individual behaviour to consider emotional processes in social psychology models. An individual may generate forward-looking emotions towards future behaviour when the future is uncertain. Past behaviour is assumed to be a potentially influential factor on behavioural intention and actual behaviour [37]. Perugini and Bagozzi [38] argued that desire can be considered as a mental state derived from the stimulation process of a certain behaviour, which in turn acts as a motivational factor to motivate individuals to carry out a specific behaviour. In the MGB, attitude (AT), subjective norm (SN), perceived behavioural control (PBC), positive anticipated emotion (PAE), negative anticipated emotion (NAE), and frequency of past behaviour (FPB) are the antecedent variables that affect behavioural intention (BI) through desire (DE) (Please see Table 1 Abbreviation for academic terms in this study). 

Several studies are employing the MGB as a theoretical framework to explain and predict individuals’ behavioural intention or actual behaviour in leisure and tourism [39,40,41,42]. There are two ways to enhance the theoretical impact of MGB in certain contexts, namely the integration of new critical constructs and the alteration of existing paths in the original theory [36,40]. Considering the unique context of smog pollution, it is therefore imperative to integrate new constructs into the MGB to understand the formative mechanism of an individual’s behaviour comprehensively [36].

### 2.6. Hypotheses Development

#### 2.6.1. Relationships among the MGB

The MGB is an advanced model developed from TPB: the antecedent variables (i.e., AT, SN, PBC) in TPB are proven to affect behavioural intention directly. In the MGB, AT refers to an individual’s overall appraisal of performing a certain behaviour; SN reflects individual perceived social pressure from significant others on whether to carry out a particular behaviour; and PBC examines an individual’s overall perception of the ease or difficulty in performing a specific behaviour [43]. These three antecedent variables fortify one’s desire to conduct the specific behaviour at first and subsequently affect behavioural intention [36]. In the MGB, desire acts as a driver of behavioural intention, while an individual’s dynamic self-regulatory process could also be appraised through two anticipated emotions in the MGB. Specifically, the two anticipated emotions could examine an individual’s success or failure to reach a goal, and positively anticipated emotion reflects expected compensation through achieving a goal, whereas negatively anticipated emotion reflects expected failure in achieving a goal, which can predict one’s desire to perform a specific behaviour and his/her behavioural intention [44]. Past behaviour is considered as the continuation of habit; prior studies showed that if an individual conducts a specific behaviour as habits, such behaviour may further influence his/her desire and behavioural intention [36]. Thus, based on the MGB model, we put forth our hypotheses:

**Hypothesis** **1** **(H1).***AT is positively correlated with desire*.

**Hypothesis** **2** **(H2).***SN is positively correlated with desire*.

**Hypothesis** **3** **(H3).***PBC is positively correlated with desire*.

**Hypothesis** **4** **(H4).***PBC is positively correlated with behavioural intention*.

**Hypothesis** **5** **(H5).***PAE is positively correlated with desire*.

**Hypothesis** **6** **(H6).***NAE is negatively correlated with desire*. 

**Hypothesis** **7** **(H7).***FPB is positively correlated with desire*. 

**Hypothesis** **8** **(H8).***FPB is positively correlated with behavioural intention*.

**Hypothesis** **9** **(H9).***DE is positively correlated with behavioural intention*. 

#### 2.6.2. Relationship between the Mass-Media Effect and Perception of Smog 

The media is an important communication channel for the public to access information, and it is also influential in affecting public perception and awareness [23,45]. Media will strongly influence the public’s priority in matters of public concern because the public will judge which topic is important based on the extent of media coverage [24]. Sun, Cang, Ruan, and Zhu [26] argued that mass-media coverage on environmental issues, especially smog pollution and its impact on air quality, will influence individual perceptions. This implies that the mass media may provide a critical access to guide the public’s awareness of environmental issues [24]. Recently, Yang and Wu [46] examined the media effect on individual’s perception of protective behaviour against smog pollution. Thus, it is assumed that the mass-media effect (MME) has a positive effect on the perception of smog in Beijing.

**Hypothesis** **10** **(H10).***MME is positively correlated with perception of smog*.

#### 2.6.3. Relationship between Mass-Media Effect, Desire, and Behavioural Intention

The correlation between mass media and public perceptions has been widely discussed in social studies [47]. The widespread prevalence of the Internet is a significant factor that changes tourist consumption patterns and their experience of tourism products [48]. Tasci and Boylu [49] found that the impact of disaster events on local tourism will continue to be strengthened by reports from the mass media. The mass media can establish specific issues as urgent public priorities through extensive and prominent news reporting, such as smog pollution issues, including the causes, consequences, and potential solutions for smog pollution [50]. Reisinger and Mavondo [51] suggested that once tourists realise a potential danger or risk at the tourism destination, their visiting desire will decline. In China’s case, the widespread smog information and related reports on a global scale have damaged China’s national image and hinder the growth of China’s inbound tourism [32]. Peng and Xiao [52] revealed that smog pollution in a tourism destination will reduce tourists’ willingness to visit the destination. This finding implies that smog pollution hinders the growth of China’s tourism. Given the media’s focus on reporting the threat of smog, visitors will have heightened awareness of the risks of their external environment when making travel-related decisions. Based on the aforementioned, this study posits the following hypotheses:

**Hypothesis** **11** **(H11).***MME is positively correlated with desire*.

**Hypothesis** **12** **(H12).***MME is positively correlated with behavioural intention*.

#### 2.6.4. Relationship among Perception of Smog, Desire, and Behavioural Intention

Psychological research suggested that an individual is likely to avoid the object when he/she holds negative cognition/perception of the object [53]. Tasci and Boylu [49] confirmed that disaster events could change individuals’ perception of tourism destinations and directly influence the tourism decisions of tourists, and this impact is continuous. Zhang, Yang, Zhang, and Zhang [54] reported that tourists in Beijing showed fewer positive emotions and more health concerns due to severe smog pollution. Specifically, tourists showed more sensitively to air pollution compared to residents. Tourists were more likely to avoid outdoor exposure when the air quality was not good. It was also revealed that tourists’ perception of health risks caused by smog may weaken their willingness to visit a destination, shorten their stay period, and even lead to their disloyalty [33]. Severe smog pollution at the destination may become a significant consideration for potential visitors when making travel plans. General growing concerns about the poor air quality in many Chinese cities are weakening China’s overall attractiveness as a tourism destination [13]. Severe smog pollution in China will diminish tourists’ experience and impede the growth of inbound tourism [16,20]. Thus, it is hypothesised that perception of smog (POS) has a negative effect on desire and behavioural intention to travel to Beijing as follows:

**Hypothesis** **13** **(H13).***POS is positively correlated with desire*.

**Hypothesis** **14** **(H14).***POS is positively correlated with behavioural intention*.

## 3. Methodology

### 3.1. Case Area

Beijing is the capital city of China and one of China’s most popular international tourism destinations owing to its long history and rich cultural tourism appeal. It is also one of the hardest-hit areas in terms of the frequency of smog pollution and the severity of particulate matter in the atmosphere [11]. Inbound tourists to Beijing have increased steadily since 2005 except for a decline in 2009 because of the 2008 World Financial Crisis. By the end of 2018, the number of Beijing inbound tourists remained at a steady level of 4.001 million. Smog pollution is widely believed to be the principal cause of the decline in the growth of inbound Beijing tourism. As confirmed by Tang, Ma, and Song [55], given a 1% increase in the number of smoggy days in Beijing, the inbound tourist arrival would decrease correspondingly by 0.25 million. Beijing is a popular inbound tourist destination in China and is representative of China’s national image. Therefore, there is a pressing need to speed up the recovery of Beijing’s inbound tourism market. There are, of course, many plausible reasons for the slow recovery of Beijing’s inbound tourism, such as exchange rate fluctuations and government intervention. However, smog pollution is an influential factor that needs to be chiefly considered, given that severe smog pollution will reduce international tourists’ intentions to visit Beijing.

### 3.2. Data Collection and Analysis

A self-administrated questionnaire was distributed to the target sample: international tourists arriving in Beijing, China. This research gives an opportunity to interview on-site international tourists in Beijing, a rarity in the current smog studies. A formal survey was conducted twice to make the sample more representative. The first survey was conducted from January to February 2018 in the Capital International Airport in an area near the Palace Museum and China’s National Museum in Beijing. The second survey was conducted from April to May 2018 at the same places. A total of 528 completed surveys from 550 international tourists were received with a total response rate of 96.0%. A total of 26 questionnaires were further eliminated because of incomplete responses or irregularities. Finally, a total of 502 questionnaires were coded and used for further analysis.

In this section, the general operationalisation of the EMGB framework to measure international tourist’s behavioural intention was suggested under the threat of smog pollution in Beijing. The questionnaire in this study included three sections. The first section measured the original MGB’s parameter using 29 items addressing AT, SN, PBC, PAE, NAE, DE, BI, and FPB. Respondents could rate their level of agreement with each statement guided by a 5-point Likert scale (1 = strongly disagree and 5 = strongly agree). The second section measured integrated variables associated with MME and POS, which were generated from the prior studies of smog pollution or climate change. Respondents also rated their level of agreement with each statement guided by a 5-point Likert scale (1 = strongly disagree and 5 = strongly agree). The last section measured respondents’ demographic information: gender, nationality, age, marriage status, education, and occupation.

In this study, R-statistic was applied to perform descriptive statistics, confirmatory factor analysis (CFA), and structural equation modelling (SEM). Specifically, this study used CFA to test the EMGB measurement model’s appropriateness and determine the internal consistency of the items and its construct validity. Finally, the structural relationships between the EMGB variables were verified by SEM.

## 4. Result

### 4.1. Demographic Characteristics

The demographic characteristics of the respondents are shown in Table 2. The proportion of male respondents (64.74%) is higher than female (35.26%). The majority of respondents are aged between 20–29 (45.02%) and 30–39 (19.12%). University or higher graduates are predominant (84.66%). Most respondents are single (61.16%). Europeans comprised 22.31% of the sample, followed by tourists from the U.S. (20.32%), Japan (11.95%), and Korea (11.16%). The proportion distribution of the main countries of origin is congruent with the statistical reports of Beijing Municipal Bureau of Statistics (2017) [56]. A total of 26.70% of the respondents are students, followed by experts or technicians (22.31%).

### 4.2. Measurement Model

First, this study performed CFA to estimate the measurement model for all the EMGB variables [57]. Mardia’s standardised coefficient of this measurement model suggests that the data violated the assumption of multivariate normality; Mardia’s standardised coefficient was 34.296 (greater than the criteria of 5) [58]. This suggested that the robust maximum likelihood method should be used in order to provide a more robust and valid standard error and other fit indexes [58]. The proposed measurement model fitted the data well, with the goodness of fit shown in Table 3 (NFI = 0.906, NNFI = 0.936, CFI = 0.944, RMSEA = 0.051).

The measurement model satisfied the convergent validity, with all factor loadings (see Table 4) exceeded the criteria of 0.5 [57]. Moreover, each construct in the EMGB suggested a sufficient reliability level, as all values of Cronbach’s alpha (0.883 to 0.950) were greater than the criteria of 0.7 [61].

Construct validity was verified via convergent and discriminant validity. All average variance extracted (AVE) and composite reliability (CR) in the measurement model for the EMGB were greater than the criteria of 0.5 and 0.7, respectively [60]. This indicated that the measurement model for the EMGB has a sufficient level of convergent validity. Also, all AVEs of each construct were greater than the squared correlation (see Table 5), which demonstrates satisfactory discriminant validity.

### 4.3. Hypothesis Testing

The structural model of the EMGB was found to fit the data well with better goodness of fit (NFI = 0.898, NNFI = 0.931, CFI = 0.938, RMSEA = 0.053). The results of the EMGB are shown in Figure 3. Six predictor variables (AT (β_AT__→__DE_ = 0.170, t = 2.316, *p* < 0.05), SN (β_SN__→__DE_ = 0.146, t = 2.502, *p* < 0.05), PBC (β_PBC__→__DE_ = 0.137, t = 3.016, *p* < 0.01), PAE (β_PAE__→__DE_ = 0.214, t = 3.720, *p* < 0.001), NAE (β_NAE__→__DE_ = 0.148, t = 3.484, *p* < 0.001), and FPB (β_FPB__→__DE_ = 0.094, t = 2.365, *p* < 0.05) were positively associated with desire to visit Beijing, while POS (β_POS__→__DE_ = −0.187, t = −3.462, *p* < 0.001) was negatively associated with desire to visit Beijing, supporting H1, H2, H3, H4, H5, and H7. However, MME (β_MME__→__DE_ = 0.082, t = 1.605) was not statistically significant to predict the desire to visit Beijing, thereby rejecting H11.

The relationships among PBC, MME, desire, and BI were found to be positive and significant (β_PBC__→__BI_ = 0.067, t = 2.128, *p* < 0.05; β_DE__→__BI_ = 0.759, t = 27.844, *p* < 0.001; β_MME__→__BI_ = −0.101, t = −2.737, *p* < 0.01), supporting H6, H9, H12. However, the FPB and POS (β_FPB__→__BI_ = −0.033, t = −0.789; β_POS__→__BI_ = −0.055, t = −1.548) were not statistically significant to predict behavioural intention to visit Beijing, thereby rejecting both H8 and H14.

It is interesting to note that MME could affect BI through two paths: one is a direct path; the other affects BI indirectly through POS and DE sequentially. The finding indicated that DE is a strong drive to transform individual’s perception into behavioural intention. Also, MME and POS are closely related to international tourists’ behavioural intention for Beijing travel.

## 5. Discussion

### 5.1. Discussion

Although smog pollution has been considered as a critical factor responsible for the decline of inbound tourism in China [14,16], research remains scarce regarding international tourists’ behavioural intention when facing smog pollution in China. This study developed an extended MGB with the integration of mass-media effect and perception of smog to uncover the conscious, emotional, and deliberate travel decision-making process of international tourists when faced with smog pollution. This study can provide insight for the Chinese government and related departments to manage the negative impact caused by smog pollution, which could be seen as an essential basis for the government to formulate a rational urban tourism-development strategy.

Findings showed the constructs of the EMGB could effectively predict international tourists’ intentions to travel to Beijing when faced with smog pollution. In all antecedent variables of the EMGB, desire is the most crucial latent variable for behavioural intention, which also acts as a sufficient impetus for behavioural intention formation. Among determinants of desire, positive anticipated emotion was proven to be essential, which implies that individuals prefer the destination that can bring them positive emotions. This could be explained by considering that tourism experiences often include satisfying and pleasurable emotions [62], which would not be discovered when employing the TPB. Besides, the smog perception of international tourists is another main predictor of desire. This finding revealed that international tourists are more likely to decrease their willingness to travel for Beijing due to smog pollution rather than other cognitive factors of Beijing tourism, such as attitude, subjective norms, and perceived behavioural control. This might be attributable to the fact that environmental quality is always a prime component of the tourism destination’s attractiveness. Beerli and Martín [63] argued that climate and weather would directly affect tourists’ comfort perception, physical health, and tourist satisfaction. In the current study, tourists’ choices could be affected by air quality, as this is highly visible and occurs seasonally (usually in spring and winter). Air quality could, therefore, be an important consideration affecting the macroscopic environment of tourism development, seasonality, and tourism activities. This finding was also not be discovered in the original MGB.

Notably, no significant causal relationships exist between mass-media effect, frequency of past behaviour, and behavioural intention. However, the mass-media effect and frequency of past behaviour can affect behavioural intention through desire. This implies that desire is a decisive, mediated variable between the endmost constructs of the model and behavioural intention. Also, mass media affects behavioural intention through different paths. There is indirect impact on behavioural intention through the perception of smog and desire. In the indirect path, it is noteworthy that the mass-media effect has a positive and significant perception of smog. This implies that international tourists perceive smog as harmful to a certain extent. They understand that the smog in Beijing is severe; they are aware of the smog’s harm to society; and they worry that the smog will damage their health during their stay [11]. The public (including tourists) mainly get information about smog pollution in Beijing through the mass media, and mass media could promote tourist perception of smog pollution. A negative perception would affect tourists’ desire to visit Beijing, which is not conducive to establishing tourist behavioural intentions. In a word, the mass-media effect is significant in the decision-making process of international tourists towards travel to Beijing.

### 5.2. Theoretical Implications

First, the result of EMGB implies that new variables (i.e., mass-media effect and perception of smog) could be added to the original MGB. This finding is consistent with Ajzen’s [43] study, which suggested that the psychosocial model is open to change mainly through two ways: one is considering subject-related factors, and the other is changing relationships among latent variables if they can explain a substantial proportion of the total variance of behavioural intention. Additionally, the results supported the findings of existing studies [39,40,41] in which necessary variables can be added into the original MGB to better understand human behaviour. The EMGB contributes to avoiding possible misspecification, including omitting important variables or the consideration of unimportant variables in the model. In other words, the two new constructs (mass-media effect and perception of smog) were effectively considered in understanding the intricate mechanism that forms behavioural intention to travel for Beijing when faced with smog pollution.

Second, the current study emphasized the role of mass-media coverage in individual perception formation and decision-making process regarding to health issues. In prior studies, mass media was proven to accelerate the spread of such risk events, like disease (e.g., SARS, H1N1, COVID-19), natural disasters (e.g., earthquake, flood, tsunami), and terrorism, which will further shape individuals’ risk perception or negative image of a destination [49]. This study emphasized the mass-media effect towards perception of smog, which broadened the research scope of individual perception to risk event. Findings revealed that the mass-media effect enhances tourist perception of smog, and the perception of smog further decreases desire and behavioural intention to travel for Beijing sequentially. This finding implemented evidence to emphasize the role of mass media in the decision-making process. As Becken, Jin, Zhang, and Gao [13] reported, there is correlation between the air quality of China and a tourist’s intention to visit: in particular, when the potential tourist feels the air quality of China is good, his/her willingness to visit will correspondingly increase as well. The current study implied that mass media is a vital medium that may intensify individual’s negative perception towards destinations and, therefore, weaken their subsequence travel decisions.

### 5.3. Practical Implications

Practical implications were also provided by analysis findings. As revealed in the study, mass media can strengthen the perception of smog of international tourists. Therefore, in order to minimize the negative impact of air pollution mass-media coverage on inbound tourism, it is priority for Chinese government to take efforts to reduce air pollution. That is because mass-media coverage is a main source for the international community to be educated about China’s environmental issues. For example, since 2013, whenever a severe smog episode occurred in China, the number of media reports has exponentially increased. Among them, the neighbouring countries (e.g., Korea and Japan) reported that China’s smog is responsible for the decline of air quality in their countries to some extent, while the Chinese media would complain about such reports [64]. This was especially so after several Korean media outlets said that China’s smog is the most serious atmospheric pollution in human history [63]. Obviously, these reports could damage China’s environmental image. In addition, the mass media caused a negative impact on behavioural intention, which may be attributed to the decline of Beijing’s inbound tourism. In fact, Beijing’s air quality has improved since May 2019, with the concentration of PM_2.5_ reduced to 34 μg/m^3^ [65]. However, according to statistical data, the number of international tourists’ arrivals in 2018 has not yet increased to the level of 2012. It may be helpful to take continuous efforts to stabilize China’s air quality above the guideline of WHO so that international mainstream mass media may consequentially illustrate the current air quality level of China, which could further shape public perception.

It could be implied that the international tourist’s desire to visit Beijing may be reduced because of their concern about whether Beijing’s smog may damage their health. For example, Korean and Japanese media have reported that smog increases the human mortality rate. According to Air Korea [66], outdoor classes will be suspended when the concentration of PM_2.5_ exceeded over 36 μg/m^3^ in an hour. Sales of smog-resistance products increase significantly in Korea during bouts of serious air pollution in China [66]. The mass-media coverage will undermine the established tourism image and even lead to the decline of the potential tourism demand and tourism revenue [11]. Therefore, it may be helpful to improve the tourism image of Beijing by enhancing its cultural soft power to provide tourists with a high level of tourism experience. Moreover, it is particularly important for the Chinese government to continue to improve air quality so as to reduce the impact of potential risks caused by smog on tourists’ destination choice. Building environmentally friendly facilities, such as charging facilities for new energy vehicles, can be a good idea to reduce air pollution while increasing green belt or city greenery that could absorb toxic air and concurrently create distinctive and beautiful views in the city [4]. It may also be beneficial to provide sufficient safe and health-related guarantees to the source markets, such as installing large air purifiers (towers) in polluted cities or regions and introducing smart road clean systems and bus shelter clean systems, such as installing LED screens which display daily air quality, to assure tourists of the city air quality during their stay, which may establish a positive reputation and increase the tourist revisitation rate.

### 5.4. Limitation and Future Studies

This study has some limitations that may help those conducting future studies. The first limitation is that the results may be different depending on seasonality and weather condition because the survey of the current study was conducted only for international tourists in spring and winter, which has poorer air quality than other seasons. In addition, the effect of smog as a tourism risk may have been overestimated because the study was conducted in a year other than the year when smog was most prevalent. As smog levels occur differently with different seasons and times, international tourists visiting Beijing may have different perceptions of smog accordingly [67]. Therefore, there exists the possibility that the respondents of this survey cannot fully represent the opinions of international tourists in different air quality. In further studies, survey should be conducted for international tourists taking into account the season and the time of survey period.

Second, although air pollution is one main constraint factor for China’s inbound tourism, there are still some crossed factors which may also affect international tourists’ behavioural intention, such as monetary elements of the exchange rate, world economic situation, and so on. These crossed factors can be taken into consideration in future studies.

Third, there is limitation of language bias, as the questionnaire was only translated into three languages: English, Korean, and Japanese. European countries are also main resource countries of Beijing travel, but many European people, for example, French, German, and Spanish, cannot read English very well. Additionally, many tourists from Southeast Asia do not speak English. Therefore, our questionnaire respondents from Europe and Southeast Asia were the tourists who can read English. In order to get generalized respondents, questionnaires in various languages should be made although this may also enhance the difficulty in conducting the survey.

Fourth, the sample size in the current study is not balanced, with more questionnaires spread to western tourists, which do not satisfy the requirements of a multi-group test. In further studies, a variety of survey methods can be used to equalize the sample size of eastern and western tourists. It will be a good trial for future research to compare the difference of decision-making processes between western tourists and eastern tourists in order to understand the sensitivity level of air pollution in different environmental/cultural atmospheres.

Finally, future researchers may focus on how to decrease the negative impact caused by mass-media effect. Understanding the specific forming mechanism of smog perception and how to respond to air pollution issues by referring to the prior studies or cases in the media would be helpful to minimize adverse impact caused by air pollution.

## 6. Conclusions

Environment sustainability is one main foci in tourism destination. The current study examined international tourists’ behavioural intention for Beijing travel under the threat of air pollution by extending MGB. Results proved that the EMGB could predict the behavioural intention of Beijing international tourists under the threat of smog pollution because the nine constructs of the EMGB significantly predicted the behavioural intention of international tourists directly or indirectly.

Specially, positive anticipated emotion generates stronger effect on desire to travel to Beijing compared to other constructs in original MGB, whereas perception of smog was proven to be negatively correlated with this desire. Findings also indicated that mass media can affect behavioural intention indirectly through perception of smog and desire. It implies that desire is a crucial factor to intention formation, and mass-media effect and perception of smog also closely related with international tourist’s behavioural intention. Overall, this study emphasized the crucial role of mass media in the formation of perception and behavioural intention under the guidance of MGB.

## Figures and Tables

**Figure 1 ijerph-18-07262-f001:**
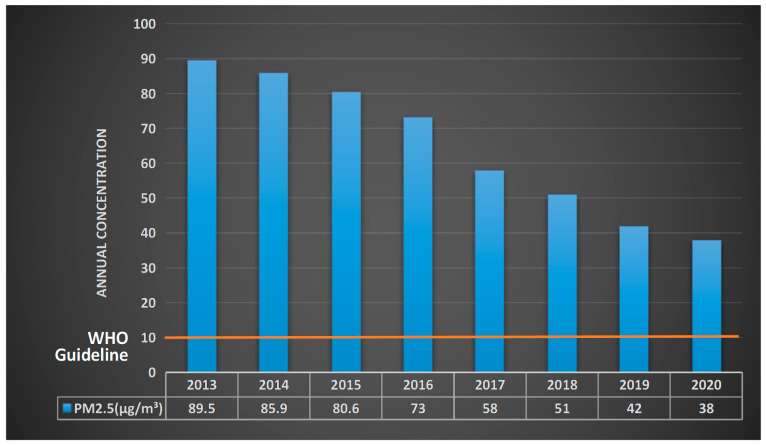
PM_2.5_ annual concentration of China’s air from 2013 to 2020.

**Figure 2 ijerph-18-07262-f002:**
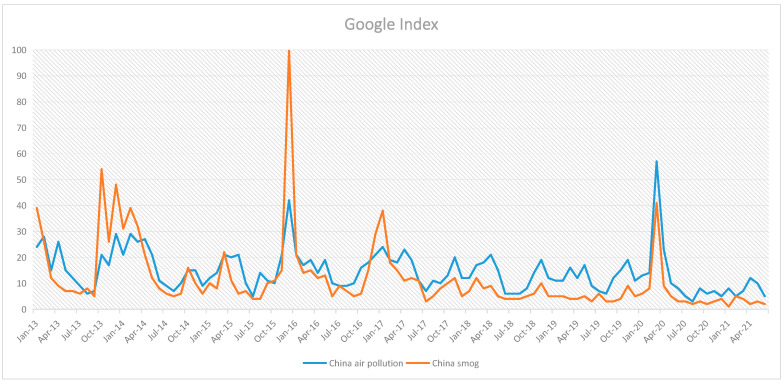
Results of Google Trends.

**Figure 3 ijerph-18-07262-f003:**
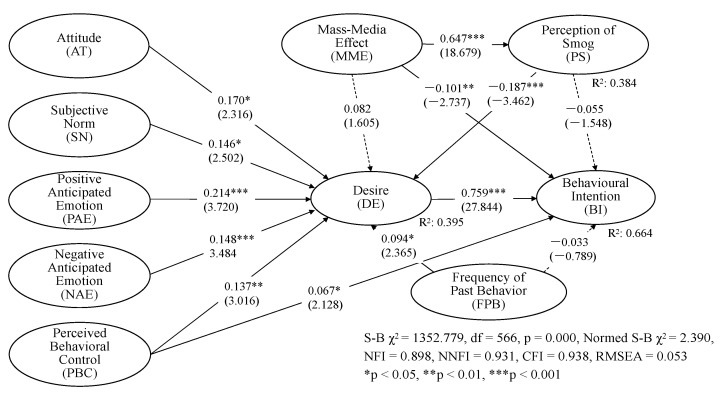
Results of EMGB. Note: the numbers in the parenthesis indicate t-value.

**Table 1 ijerph-18-07262-t001:** Abbreviation for academic terms in this study.

Academic Terms	Abbreviation
Attitude	AT
Subjective norm	SN
Perceived behavioural control	PBC
Positive anticipated emotion	PAE
Negative anticipated emotion	NAE
Frequency of past behaviour	FPB
Mass-media effect	MME
Perception of smog	POS
Desire	DE
Behavioural intention	BI

**Table 2 ijerph-18-07262-t002:** Demographic characteristics of the respondents. (N = 502).

Characteristic	N (%)	Characteristic	N (%)
**Gender**		**Marital status**	
Male	325 (64.74)	Single	307 (61.16)
Female	177 (35.26)	Married	175 (34.86)
		Other	20 (3.98)
**Education level**		**Age**	
Less than high school	28 (5.58)	Under 20	26 (5.18)
College	49 (9.76)	20–29	226 (45.02)
University	273 (54.38)	30–39	96 (19.12)
Graduate school	152 (30.28)	40–49	63 (12.55)
		50–59	52 (10.36)
		Over 60	39 (7.77)
**Nationality**		**Occupation**	
Southeast Asia	19 (3.78)	Expert or technician	112 (22.31)
U.S.	102 (20.32)	Self-employed	71 (14.14)
Russia	51 (10.16)	Service	24 (4.78)
Europe	112 (22.31)	Office staff	66 (13.15)
Australia	20 (3.98)	Civil servant	7 (1.39)
India	25 (4.98)	Military	5 (1.00)
Africa	20 (3.98)	Student	134 (26.70)
Canada	32 (6.37)	Housewife	15 (2.99)
Korea	56 (11.16)	Freelance	33 (6.57)
Japan	60 (11.95)	Retired	9 (1.79)
Other	5 (1.00)	Others	26 (5.18)

**Table 3 ijerph-18-07262-t003:** Goodness-of-fit indices of measurement model of EMGB.

Structural Model	S-B χ^2^	df	Normed S-B χ^2^	NFI	NNFI	CFI	RMSEA
Fit indices	1218.210	524	2.325	0.906	0.936	0.944	0.051
Suggested value			≤3	≥0.9	≥0.9	≥0.9	≤0.08

Note: Suggested values were based on Bearden, Sharma, and Teel [59] and Hair, Black, Babin, Anderson, and Tatham [60].

**Table 4 ijerph-18-07262-t004:** Reliability and confirmatory factor analysis.

Constructs	Factor Loading	t-Value	Cronbach’s Alpha
Attitude (AT)			0.883
I think traveling to Beijing is valuable.	0.784	30.897	
I think traveling to Beijing is positive.	0.814	38.124	
I think traveling to Beijing is attractive.	0.818	38.914	
I think traveling to Beijing is beneficial.	0.821	44.591	
Subjective Norm (SN)			0.873
People who are important to me recommend that I travel to Beijing.	0.727	26.389	
_____________ agree that I should travel to Beijing.	0.789	30.984	
_____________ understand I should travel to Beijing.	0.793	37.813	
_____________ support my decision to travel to Beijing.	0.866	46.114	
Perceived Behavioural Control (PBC)			0.874
I have enough financial resources to travel to Beijing.	0.745	31.872	
I am confident that if I want, I can travel to Beijing.	0.850	49.729	
I have enough time to travel to Beijing.	0.920	60.657	
Positive Anticipate Emotion (PAE)			0.918
If I travel to Beijing, I will be excited.	0.843	45.273	
If I travel to Beijing, I will be happy.	0.843	29.309	
If I travel to Beijing, I will be satisfied.	0.868	55.962	
If I travel to Beijing, I will be glad.	0.885	59.901	
Negative Anticipate Emotion (NAE)			0.884
If I cannot travel to Beijing, I will be angry.	0.772	28.596	
If I cannot travel to Beijing, I will be disappointed.	0.807	32.008	
If I cannot travel to Beijing, I will be worried.	0.811	37.658	
If I cannot travel to Beijing, I will be sad.	0.851	42.584	
Mass-Media Effect (MME)			0.939
The mass media (TV, news, internet) notifies of the risk of smog in Beijing.	0.873	66.767	
The mass media notifies of the negative impact of smog on modern society in Beijing.	0.886	58.850	
The mass media notifies of the severity of smog in Beijing.	0.896	80.660	
The mass media notifies of the negative impact of smog on human health in Beijing.	0.911	89.946	
Perception of Smog (POS)			0.903
I recognize that smog is progressing drastically in Beijing.	0.785	36.356	
I sense the negative impact of smog on modern society in Beijing.	0.836	44.202	
I think the recent drastic smog in Beijing is threatening.	0.840	46.678	
I sense the negative impact of smog on human health in Beijing.	0.892	63.137	
Desire (DE)			0.950
I am eager to travel to Beijing again in the near future.	0.895	70.068	
I am enthusiastic about traveling to Beijing again in the near future.	0.903	84.023	
I would like to travel to Beijing again in the near future.	0.917	92.086	
I hope to travel to Beijing again in the near future.	0.918	96.368	
Behavioural Intention (BI)			0.950
I am willing to invest money and time to travel to Beijing again in the near future.	0.877	65.324	
I will make an effort to travel to Beijing again in the near future.	0.914	89.791	
I plan to travel to Beijing again in the near future.	0.921	96.721	
I have an intention to travel to Beijing again in the near future.	0.924	114.610	

Note: Mardia’s standardized coefficient = 34.296. All standardized factor loadings are significant at *p* < 0.001. Suggested values were based on Bearden et al. [59] and Hair et al. [60].

**Table 5 ijerph-18-07262-t005:** Result of measurement model. (N = 502).

	AT	SN	PBC	PAE	NAE	MME	POS	DE	BI
AT	0.655								
SN	0.510 ^a^ (0.260) ^b^	0.633							
PBC	0.332 (0.110)	0.465 (0.216)	0.708						
PAE	0.618 (0.382)	0.546 (0.298)	0.364 (0.133)	0.740					
NAE	0.380 (0.144)	0.353 (0.125)	0.129 (0.017)	0.373 (0.139)	0.657				
MME	−0.035 (0.001)	−0.128 (0.016)	−0.174 (0.030)	−0.016 (0.000)	0.046 (0.002)	0.795			
POS	−0.099 (0.010)	−0.135 (0.018)	−0.110 (0.012)	0.043 (0.002)	−0.069 (0.005)	0.620 (0.384)	0.704		
DE	0.478 (0.228)	0.478 (0.229)	0.366 (0.134)	0.497 (0.247)	0.370 (0.137)	−0.100 (0.010)	−0.196 (0.038)	0.825	
BI	0.439 (0.192)	0.416 (0.173)	0.368 (0.136)	0.349 (0.122)	0.349 (0.122)	−0.215 (0.046)	−0.271 (0.074)	0.800 * (0.639)	0.827
CR	0.884	0.872	0.878	0.919	0.885	0.939	0.905	0.950	0.950

Note: CR = composite reliability; AVE (average variance extracted) is on the diagonal line. ^a^ Correlations between variables are below the diagonal. ^b^ Squared correlations are within the parentheses. * Pairs of constructs having highest correlations.

## Data Availability

Data sharing not applicable.

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
