# Peer review of "Understanding Tourist Behavioural Intention When Faced with Smog Pollution: Focus on International Tourists to Beijing"

_ijerph, 2021, doi:10.3390/ijerph18147262_

Round 1

Reviewer 1 Report

1. Although there is disagreement about writing articles in the first person, I still follow the classical school that provides for writing in the impersonal form, that is, in the third person. I suggest revising and correcting the entire article.
2. The authors based their work only on the event that took place in China in 2013.
3. Also, the Google search on air quality (smog pollution) should not be the only one; the authors should reference the work in official data, scientific papers, and other reliable sources.
4. The literature review brings more than known concepts, and a systematic review of literature on the subject was not performed. Are there other studies? Have there been similar results in other countries, even with diverse natural resources impacting tourist behavior? A detailed systematic literature review should be done.
5. The questionnaires were applied in 2018, a year other than the year the authors mentioned having a significant smog occurrence; how do you correlate tourist behavior in a year other than the one that occurred as an element that would drive them away?
6. The crossed factors for the behavioral analysis of the tourist were not taken into consideration, such as monetary elements of the exchange rate, world economic situation, etc.
7. The conclusions made by the authors do not agree with what could be extracted from the questionnaires.

Author Response

Response to Reviewer 1 Comments

Point 1: Although there is disagreement about writing articles in the first person, I still follow the classical school that provides for writing in the impersonal form, that is, in the third person. I suggest revising and correcting the entire article.

Response 1: Thanks for your comments. We have revised and correcting the first person to impersonal form as you suggested.

Point 2: The authors based their work only on the event that took place in China in 2013.

Response 2: Thanks for your comments. We have provided more evidences to support air pollution is a main constrain factor for China’s inbound tourism.

The current widespread air pollution issue in China was mainly responsible for the decline in inbound tourist market. Therefore, the State Council of China announced the Air Pollution Prevention and Control Plan on 10 September, 2013 with the purpose to control polluted air emissions and further improve air quality. Until 2017, the air quality of China has improved a lot compared to that of 2012. But the concentration of particulate matter in Beijing, namely PM2.5 (73μg/m³) and PM10 (92μg/m³), continues to exceed WHO guidelines and poses substantial threats to human health. It implies that smog pollution of China seems still a constrain for international tourist arrivals.

Point 3: The Google search on air quality (smog pollution) should not be the only one; the authors should reference the work in official data, scientific papers, and other reliable sources.

Response 3: Thanks for your comments. We have provided official reports from the World Economic Forum 2019 to support air pollution has been spread in mass media, like official reports.

For example, WHO labeled air pollution in China a health crisis and according to the World Economic Forum (2019), China’s overall tourism competitiveness ranks 13th among 140 countries, whereas its environmental sustainability index ranks 120th with air pollution ranking 137th.

Added reference: Calderwood, L. U.; Soshkin, M. The travel & tourism competitiveness report 2019; World Economic Forum: 2020.

Point 4: The literature review brings more than known concepts, and a systematic review of literature on the subject was not performed. Are there other studies? Have there been similar results in other countries, even with diverse natural resources impacting tourist behavior? A detailed systematic literature review should be done.

Response 4: Thank you very much for proposing these questions. We have added a part “Tourist’s Behavioral Intention under Air Pollution” as follows.

“2.4 Tourist’s Behavioral Intention under Air Pollution

Severe smog is assumed to threat the sustainable development of tourism destination by jeopardizing its attractiveness and competitiveness[16]. Individuals mainly response to air pollution in avoidance attitude or show negative perception. For onsite tourists, Oltra and Sala (2014)[25] found that individual prefer to avoid exposure to polluted air when travelling. Li et al. (2016)[9] indicated that Chinese domestic visitors to Beijing had a strong negative perception of smog, which is likely to significantly reduce visitor satisfaction of Beijing travel. It was also revealed that tourist’s perception of health risk caused by smog may shorten his/her stay period, and even lead to his/her disloyalty [31,32]. For potential tourists, air pollution was proved to be a crucial factor that will suppress China’s international tourism. Specifically, Cheung and Law (2001)[33] reported that air pollution in Hong Kong was a consideration in travel decision-making process although it stays at an acceptable level, furthermore, compared with western tourists, Asian tourists paid more attention to air quality when selecting destinations. Becken, Jin, Zhang, and Gao (2017)[11] revealed that prospective international travelers’ cognitive and affective perceptions about air quality negatively influenced China’s destination image and tourists’ corresponding visit intentions. Xu and Reed (2017)[14] assessed the impact of pollution on potential international tourists and reported that perceived pollution can impede inbound tourism. Recently, Ruan, Kang and Song (2020)[34] pointed out that international tourists will avoid travelling to China if they perceived severity of air pollution in China.

Point 5: The questionnaires were applied in 2018, a year other than the year the authors mentioned having a significant smog occurrence; how do you correlate tourist behavior in a year other than the one that occurred as an element that would drive them away?

Response 5: Thanks for your comment. Here are the responses for this question:

Because Smog levels in 2018 were still severe, if not the best it seems that the validity of current study is still confirmed. For instance, the level of air pollution was still bad to limit the development of inbound tourism in China and there was no significant change to affect in tourists’ behavior in 2018.

Specifically, the Chinese government has taken a series of measures to reduce air pollution since 2011, but China’s air quality does not meet WHO guidelines until 2018. It indicates that the risk of air pollution is strongly acting as one of the only factors limiting the arrival of international tourists (Tang et al., 2019; Ruan, Kang, & Song, 2020).

Reference:

Tang, C.C.; Ma, L.; Song, Y.C. Do fog and haze affect Beijing inbound tourism? An empirical study based on panel data. Journal of Arid Land Resource and Environment 2017, 31, 192-197.

Ruan, W.; Kang, S.; Song, H.J. Applying protection motivation theory to understand international tourists behavioural intentions under the threat of air pollution: A case of Beijing. Current Issues in Tourism 2020, 23, 2027-2041, doi:10.1080/13683500.2020.1743242.

Point 6: The crossed factors for the behavioral analysis of the tourist were not taken into consideration, such as monetary elements of the exchange rate, world economic situation, etc.

Response 6: Thanks for your comments. We think it can be take into consideration in future studies, so we added this opinion in the limitation part.

“Second, although air pollution is one main constrain factor for China’s inbound tourism, there are still some crossed factors which may also affect international tourist’s behavioral intention, such as monetary elements of the exchange rate, world economic situation, and so on. These crossed factors can be taken into consideration in future studies.”

Point 7: The conclusions made by the authors do not agree with what could be extracted from the questionnaires.

Response 7: Thanks for your comments. We have recognized the conclusion part and revised some descriptions follows your comments.

“It is interesting to note that MME could affect BI through two paths: one is direct path to affect BI; the other is affect BI indirectly through POS and DE. While POS could affect BI indirectly through DE. This finding indicated that DE is a strong driven to transform individual’s perception to intention. Also, MME and POS are closely related to international tourists’ behavioural intention for Beijing travel.”

“Among determinants of desire, positive anticipated emotion were proved to be essential, which implies that individual prefer to the destination that can bring them with positive emotions.”

“This finding revealed that international tourists are more likely to decrease their willingness to travel for Beijing due to smog pollution rather than other cognitive factors of Beijing tourism, such as attitude, subjective norms, and perceived behavioural control.”

Reviewer 2 Report

This study investigated the behavioral intentions of international tourists under smog pollution. The topic is important and interesting. But there are some problems listed as follows. 

  1. It’s better to provide an abbreviation table.
  2. Section 1 and 2 should be combined and simplified. It’s too redundant.
  3. The table should be revised to standard three-line table.
  4. The discussion section and conclusion section should be separate.
  5. English should be improved.

Author Response

Point 1: It’s better to provide an abbreviation table.

Response 1: Thanks for your comments. This table has been added in literature review part.

Table 1. Abbreviation for academic terms in this study

Academic terms

abbreviation

Attitude

AT

Subjective norm

SN

Perceived behavioural control

PBC

Positive anticipated emotion

PAE

Negative anticipated emotion

NAE

Frequency of past behaviour

FPB

Mass media effect

MME

Perception of smog

POS

Desire

DE

Behavioural intention

BI

Point 2: Section 1 and 2 should be combined and simplified. It’s too redundant.

Response 2: Thanks for your comment. We have recognized section 1 and 2 as follows.

“Environmental quality has gained increasing recognition for its ability to enhance a destination’s competitive advantage. It is also important as a reference index for destination sustainability as the environment plays a prime consideration in an individual’s decision to travel [1-3]. Smog pollution is a prominent influence on human activity and behaviour because of its higher visibility and severity compared to other types of environmental pollution such as water pollution, improper waste management and soil pollution [4]. For some developing countries, the deteriorating air quality has become the main reason for decreasing number of tourist arrivals and tourism revenue, adversely affecting the economies of these countries[5]. For example, China’s attractiveness as a tourist destination tourism was greatly weakened in 2013 due to severe “smog” (“wumai” in Chinese)[6]; international tourist arrivals to Beijing totalled 26.29 million in 2013, down 3.3% from the previous year [7]. The current widespread air pollution issue in China was mainly responsible for the decline in inbound tourist market [8]. Therefore, the State Council of China announced the Air Pollution Prevention and Control Plan on 10 September, 2013 with the purpose to control polluted air emissions and further improve air quality. Until 2017, the air quality of China has improved a lot compared to that of 2012. But the concentration of particulate matter in Beijing, namely PM2.5 (73μg/m³) and PM10 (92μg/m³), continues to exceed WHO guidelines and poses substantial threats to human health. It implies that smog pollution of China seems still constrain for international tourist arrivals.

Point 3: The table should be revised to standard three-line table.

Response 3: Thanks for your comments. We have revised all tables in the manuscript to standard three-line form.

Point 4: The discussion section and conclusion section should be separate.

Response 4: Thanks for your comments. We have added conclusion part following discussion.

“6. Conclusion

Environment sustainability is one main foci in tourism destination. The current study examined international tourists’ behavioural intention for Beijing travel under the threat of air pollution by extending MGB. Results proved that the EMGB could predict the behavioural intention of Beijing international tourists under the threat of smog pollution because the nine constructs of the EMGB significantly predicted the behavioural intention of international tourists directly or indirectly.

Specially, positive anticipated emotion generates stronger effect on desire to travel for Beijing compared to other constructs in original MGB, whereas perception of smog was proved to be negatively correlated with this desire. Findings also indicated that mass media can affect behavioural intention indirectly through perception of smog and desire. It implies that desire is a crucial factor to intention formation, and mass media effect and perception of smog also closely related with international tourist’s behavioural intention. Overall, this study emphasized the crucial role of mass media in the formation of perception and behavioural intention under the guidance of MGB.”

Point 5: English should be improved.

Response 5: Thanks for your comments. We did proofread by native speaker as you suggested.

Reviewer 3 Report

Dear Authors,

I've read your manuscript on inbound tourism and smog in Beijing with interest. The manuscript is well structured and present a robust research with international tourist on site. The research questions are valid and up-to-date. For data gatherig surveys were distributed in two different time-slots which reinfroces the robustness of the study. Theoretical and practical implications are also discussed. Still, I have a few ethical concerns:

  • smog and bad air quality in Beijing is a fact whether mass media communicates it or not. It is not the mass media communication that should be changed but the smog should be reduced
  • I strongly recommend you not to suggest that Chinese government should control mass media on whatever topic. I am not located in China, but I think that mass media needs to be free of government control in any country.
  • it is not acceptable for me to read in a scientific article that the government should control mass media.
  • I also suggest you to reformulate this sentence: "due to the appearances of Asian are similar, which increases the difficulty in conducting survey to eastern tourists". This sentence is minimising coultural differences and identifies people according to their apparance...It is surprising to me to read from "Asian" researchers that Asians look all the same...which is just not true.

Author Response

Point 1: Smog and bad air quality in Beijing is a fact whether mass media communicates it or not. It is not the mass media communication that should be changed but the smog should be reduced. I strongly recommend you not to suggest that Chinese government should control mass media on whatever topic. I am not located in China, but I think that mass media needs to be free of government control in any country. It is not acceptable for me to read in a scientific article that the government should control mass media.

Response 1: Thank you very much for proposing these questions. We have rewritten these opinions from an objective perspective as follows.

“As revealed in the study that mass media can strengthen the perception of smog of international tourists. Therefore, in order to minimize the negative impact of air pollution mass media coverage on inbound tourism, it is priority for Chinese government to take efforts to reduce air pollution. That is because mass media coverage is a main source for international community to know China’s environmental issues.”

“It may be helpful to take continuous efforts to stable China’s air quality above the guideline of WHO, so that international mainstream mass media may consequentially illustrate the current air quality level of China, which could further shape public perception.”

Point 2:  I also suggest you to reformulate this sentence: "due to the appearances of Asian are similar, which increases the difficulty in conducting survey to eastern tourists". This sentence is minimizing cultural differences and identifies people according to their appearance...It is surprising to me to read from "Asian" researchers that Asians look all the same...which is just not true.

Response 2: Thanks for your comments. We have deleted this sentence, and emphasized the importance to equalize the sample size of eastern and western tourists in future studies.

Round 2

Reviewer 1 Report

Despite the author's diligence in making corrections based on the reviewers' suggestions, I believe that they made the entire methodology of the paper based on a single year (2013) and, after recommendations, extended it to 2017 still compounds the work.
A longer time frame has to be considered to evaluate pollution, primarily due to sensitive synergistic actions from emissions from other countries, natural emissions, climate change, etc.
Therefore, I maintain the suggestion to reject the article.

Author Response

Response 1: Thanks for your comments. We largely agree with the reviewers’ opinion that this paper should be much revised because this study was not conducted in the year of the highest smog level. Therefore, we added some evidences to support air pollution was still sever during this survey.

“Therefore, the State Council of China announced the Air Pollution Prevention and Control Plan on 10 September, 2013 with the purpose to control polluted air emissions and further improve air quality within 5 years. Until 2017, the air quality of China has improved a lot compared to that of 2012. But the annual concentration of particulate matter in Beijing, namely PM2.5 (58μg/m³) and PM10 (84μg/m³), continues to exceed WHO guidelines (PM2.5: 10μg/m³; PM10: 20μg/m³) and poses substantial threats to human health (See Figure 1). Although air quality in China has been increasing with Chinese government continuous efforts, there is still a long way to go as the improvement of China’s environment quality is not stable. Thus, smog pollution in China seems still constrain for international tourist arrivals.”

Figure 1. PM2.5 Annual Concentration of China Air from 2013 to 2020.

“Among enormous reports, a recent study in the lancet estimated that in 2017, 1.24 million people died from air pollution in China. This study result was cited by news of U.S. Embassy in Georgia to report air pollution in China harms its citizens and even the world.”

Figure 2. Results of Google Trends.

And the contents of this study were emphasized on the limitations of the study. Although our research level is not perfect, we ask for your consideration. The specific revised parts are as follows.

“The first limitation is that the results may be different depending on seasonality and weather condition, because the survey of current study was conducted only for inter-national tourists in spring and winter, which has poor air quality than other seasons. In addition, the effect of smog as a tourism risk may have been overestimated because the study was conducted in a year other than the year when smog was most prevalent. As smog levels occur differently with different seasons and times, international tourists visiting Beijing may have different perceptions of smog accordingly [68]. Therefore, there exists the possibility that the respondents of this survey cannot fully represent the opinions of international tourists in different air quality. In further studies, survey should be conducted for international tourists taking into account the season and the time of survey period.”

Added reference:

Yin, P.; Brauer, M.; Cohen, A.J.; Wang, H.; Li, J.; Burnett, R.T.; Stanaway, J.D.; Causey, K.; Larson, S.; Godwin, W., et al. The effect of air pollution on deaths, disease burden, and life expectancy across China and its provinces, 1990–2017: an analysis for the Global Burden of Disease Study 2017. The Lancet Planetary Health 2020, 4, e386-e398, doi:10.1016/s2542-5196(20)30161-3.

Reviewer 3 Report

The manuscript has been improved and my suggestions were accepted.

Author Response

Response 1: Thank you very much for providing suggestions to this paper. Your comments are very inspiring to us! We reorganized the results part as you suggested in the rating scale.

“It is interesting to note that MME could affect BI through two paths: one is direct path; the other is affect BI indirectly through POS and DE sequentially. The finding indicated that DE is a strong driven to transform individual’s perception into behavioural intention. Also, MME and POS are closely related to international tourists’ behavioural intention for Beijing travel.”

Round 3

Reviewer 1 Report

The justifications and changes made by the authors, after the suggestions, made me change my position.
I suggest the publication of the article, with minor revisions as to the form, native language, and correction in the references, to meet the journal's standards.
Success and congratulations to the authors.

Author Response

Response to Reviewer 1 Comments

Point 1: The justifications and changes made by the authors, after the suggestions, made me change my position. I suggest the publication of the article, with minor revisions as to the form, native language, and correction in the references, to meet the journal's standards. Success and congratulations to the authors.

 Response 1: Thank you very much for providing suggestions to this paper. Your comments are very inspiring to us! We made proofread and double checked the format of the entire manuscript as your suggestions.

Details were highlight in yellow in the manuscript.